# Formation of task representations and replay in mouse medial prefrontal cortex

**Hamed Shabani[1], Hannah Muysers[2], Yuk-Hoi Yiu[1,3,4], Jonas-Frederic Sauer[2,5], Marlene Bartos[1,2,3], Christian Leibold[1,3,4]***

[1]Bernstein Center Freiburg, University of Freiburg, Freiburg im Breisgau, Germany; [2]Institute for Physiology I, Medical Faculty, University of Freiburg, Freiburg im Breisgau, Germany; [3]BrainLinks-BrainTools, University of Freiburg, Freiburg im Breisgau, Germany; [4]Faculty of Biology, University of Freiburg, Freiburg im Breisgau, Germany; [5]Systemic Neurophysiology, Center for Integrative Physiology and Molecular Medicine, Medical Faculty, Saarland University, Homburg, Germany

***For correspondence:**
christian.leibold@biologie.uni-freiburg.de

**Competing interest:** The authors declare that no competing interests exist.

## eLife Assessment

This **useful** study characterizes the evolution of medial prefrontal cortex activity during the learning of an odor-based choice task. The evidence provided is **solid**, providing quantification of functional classes of cells over the course of learning using the longitudinal calcium recordings in prefrontal cortex, and quantification of prefrontal sequences. However, the experimental design appears to provide limited evidence to support strong conclusions regarding the functional relevance of neural sequences. The study will be of interest to neuroscientists investigating learning and decision-making processes.

**Abstract** The medial prefrontal cortex (mPFC) is thought to support cognitive flexibility by forming and maintaining generalized representations of abstract tasks. The formation of these representations as well as their relation to preexisting representations of contextual or spatial information is incompletely understood. In this study, we analyzed longitudinal one-photon calcium recordings in mice performing an olfaction-guided spatial memory task over an 8-week period that included habituation, training, and sleep epochs. Our results reveal that, while a minority of neurons initially conveyed significant information about the behavior of the animal, the bulk of task-related activity only emerged after the animals reached proficient performance. Although goal arm information is robustly represented at both the single-cell and network levels both during learning and in task-proficient mice, it undergoes significant remapping throughout the learning process. Additionally, we identified the establishment of recurring sequences during learning and their replay at reward locations, with no evidence of them existing during odor sampling phase, during sleep, or before training. Conversely, during odor sampling, information about odor identity is robustly available in the rate coactivation patterns, even before animals reached task proficiency. These findings suggest that the mPFC predominantly establishes generalized task representations de novo during learning, relying only minimally on preexisting spatial representations and that sub-second neural sequences in the mPFC are more likely involved in evaluating behavioral outcomes rather than planning future actions.

## Introduction

The medial prefrontal cortex (mPFC) is generally implicated in the representation of abstract task variables (*Miller and Cohen, 2001*; *Brincat et al., 2018*; *Kaefer et al., 2020*; *Tang et al., 2023*; *El-Gaby*

*et al., 2024*) as required, for example, in rule learning (*Wallis et al., 2001*; *Goodwin et al., 2012*; *Reinert et al., 2021*). In-depth understanding of neuronal activity patterns associated with abstract tasks and their establishment during learning strongly profits from experimental approaches available in the rodent system. However, in addition to anatomical differences between primates and rodents (*Laubach et al., 2018*; *Preuss and Wise, 2022*; *Hanganu-Opatz et al., 2023*), behavioral task designs for rodents are also often affected by spatial components, which introduce correlations between task phases and spatial locations (*Sauer et al., 2022*; *Muysers et al., 2024*; *Muysers et al., 2025*). On the one hand, this interrelation between space and task complicates interpretation of the results; on the other hand, it, however, also provides an opportunity to study potential effects of spatial schemas to task learning, as hypothesized from human fMRI data (*Zheng et al., 2021*). Unlike in the hippocampus, where CA1 place cells undergo global remapping in different contexts, PFC representations have been found to remain stable across environments, allowing for task generalization (*Tang et al., 2023*; *Muysers et al., 2024*). This stability suggests that new experiences do not completely overwrite PFC representations but instead integrate new information into existing structures, making the PFC a hub for cognitive flexibility. However, the precise nature of this reorganization remains debated. One perspective suggests that task representations in the PFC emerge de novo during learning and are distinct from preexisting spatial representations (*Symanski et al., 2022*). This view is supported by findings that task-selective sequences arise only after learning, indicating a learning-driven transformation of PFC activity. An alternative perspective, however, argues that PFC task representations build upon preexisting spatial schemas rather than forming them entirely anew (*Zheng et al., 2021*). Accordingly, the fact that PFC representations remain stable across environments may suggest that learning-related changes reflect adaptations of existing networks rather than the formation of entirely new assemblies within a circuit (*Tang et al., 2021*; *Nardin et al., 2021*). Whether PFC task representations are reorganized from prior spatial schemas or develop independently remains unresolved, highlighting a fundamental gap in our understanding of cortical learning mechanisms.

In this study, we set out to study the development of task representations on longitudinal one-photon recordings and to test whether they are based on preexisting spatial scaffolds. We reasoned that, in naive adult animals, task representations should only develop over learning, whereas the concept of space should be readily available during or even before training. We further addressed the question, whether task representations emerge as sequential assemblies in the mPFC network, or whether they are reflected as rate coactivation patterns. The former hypothesis would imply that pair correlations or activity sequences in neural populations could be interpreted as replay/preplay, whereas the latter would imply that aggregated task information would be available in population patterns and potentially provide a substrate for working memory.

Making use of a partly previously published dataset containing longitudinal mPFC recordings (*Muysers et al., 2024*; *Muysers et al., 2025*) over up to 8 weeks, encompassing habituation, training, and sleep periods, we were able to explore the development of the task code during learning. We found a small subset of goal arm-selective cells that develop general task selectivity and maintain the location of the place field. However, most of the task-related activity was observed only when the animals were proficient in task performance. Moreover, we found goal arm-selective information to be more strongly represented than general task phase information, both on the single cell and the population level. A small fraction of goal arm-selective cells showed stable firing fields throughout the recording period even before learning had started. Establishing sequence analysis on calcium transients obtained from 1p imaging, we identified the emergence of generalized task-selective sequences after learning, whereas we could find no evidence for task-related preplay during sleep or habituation to the arena. Both findings suggest that the mPFC task code is largely established during learning, with only a small amount of stable spatial tuning that is maintained from before learning.

## Results

Mice learned to perform an odor-guided navigation task (*Figure 1A*), while calcium activity was imaged in the mPFC using a miniscope (*Muysers et al., 2024*). Depending on odor identity, mice had to learn to choose either the left or the right arm and were rewarded upon the correct choice. From previous analyses (*Muysers et al., 2024*; *Muysers et al., 2025*), we expected the recorded population to display place-specific activity and asked how many of the cells were significantly spatially informative after animals became proficient in the task. We employed a complementary approach to *Muysers*

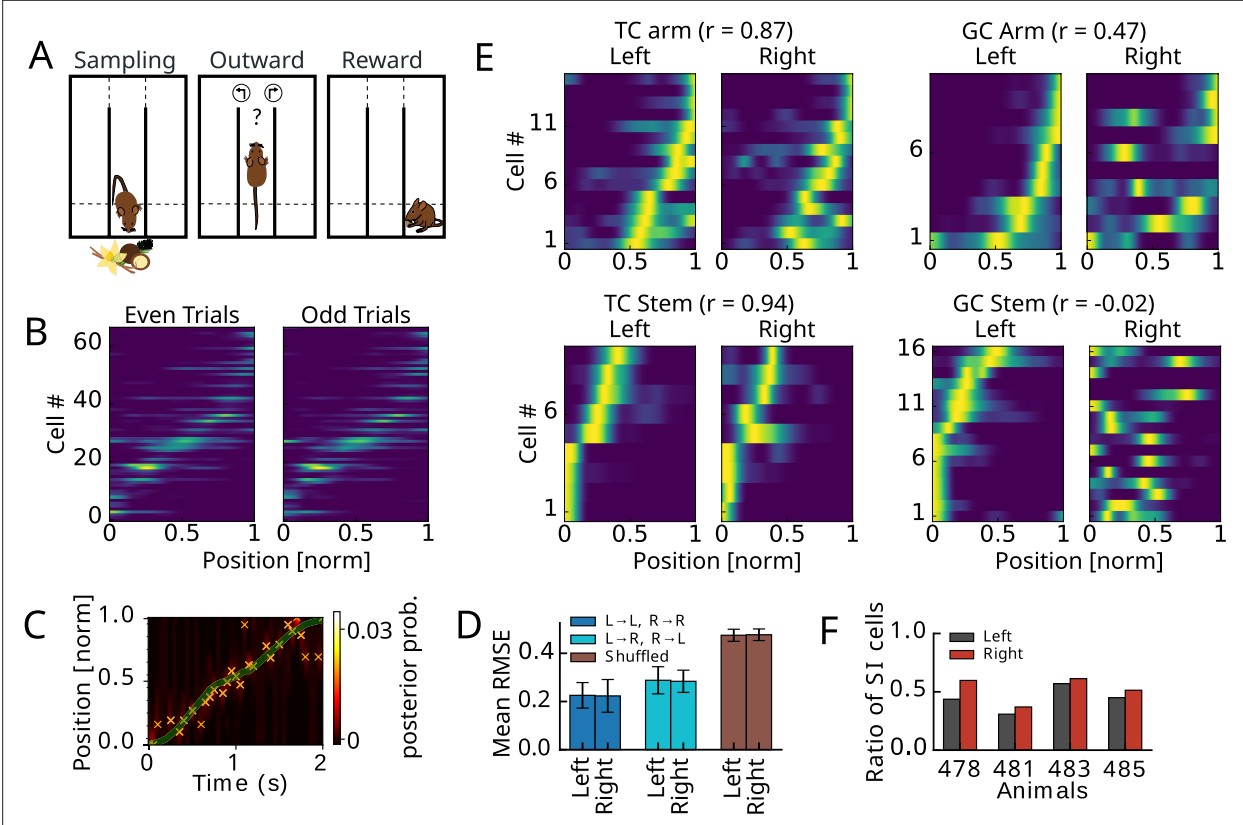

**Figure 1.** Spatially informative mPFC principal cells from proficient mice performing an odor-guided navigation task. (**A**) Mice were performing an odor-guided navigation task. We separated analysis into three behavioral phases. During Sampling, animals were exposed to an odor. During Outward, the animals were moving on the stem and one of the arms (above the horizontal dashed line). During Reward, the animals were in the reward area (below dashed line). (**B**) Activity maps (computed separately for left and right arm) obtained from one-photon calcium imaging were tested on significant spatial information (SI) content. Maps of cells with significant SI (example session) are matching between even and odd trials. (**C**) Example trial in which maximum posterior decoding (crosses) is aligned with true position (green). (**D**) Root mean square decoding errors (RMSE) for all animals and all sessions separated for left and right arm trajectories. Cross decoding RMSE (light blue) is still significantly lower than RMSE from shuffling (p<0.02; none in 50 cell index shuffles), indicating a large contribution of left-right invariant, i.e., task phase-selective activity. (**E**) Recordings from an example animal 485 separated into generalized task phase-selective cells (left) and goal arm-selective cells (right) with left and right activity maps significantly correlated or not. (**F**) About half of the cells are significantly spatially informative in all animals.

The online version of this article includes the following figure supplement(s) for figure 1:

**Figure supplement 1.** Behavioral statistics.

---

*et al., 2025* by testing for significance of spatial information content (*Skaggs et al., 1993*) (rate maps computed separately for left and right arm) with shuffle controls (see 'Methods'), and only included data from outward runs. We found that about half of the cells were significantly informative (SI) about the goal arm-specific location on the track. The place maps of SI cells, separately extracted from odd and even trials in the maze, confirmed the statistical approach (example animal in *Figure 1B*). Since in this task space is highly correlated with the task phase, we set out to distinguish place coding from task coding first, by training a Bayesian decoder on the whole population of recorded cells to predict goal arm-specific spatial position based on the recorded activity (*Figure 1C*). We found that the decoding error obtained by cross decoding (decoding left goal trajectories from the decoder trained on right goal trajectories and vice versa) had a similar magnitude as the error obtained from training and decoding on subsets of trials with the same goal arm (*Figure 1D*). Both errors were considerably and significantly lower than those obtained from shuffle controls (p<0.02 for both errors; permutation test, 50 shuffles). To further corroborate the evidence that mPFC place coding has a high contribution of task-selective activity, we differentiated between goal arm-selective and generalized task-selective cells, computed rate maps separately for left- and rightward trials of SI cells (*Figure 1E*), and computed correlation coefficients. Those SI cells for which the correlation exceeded the 95%-tile

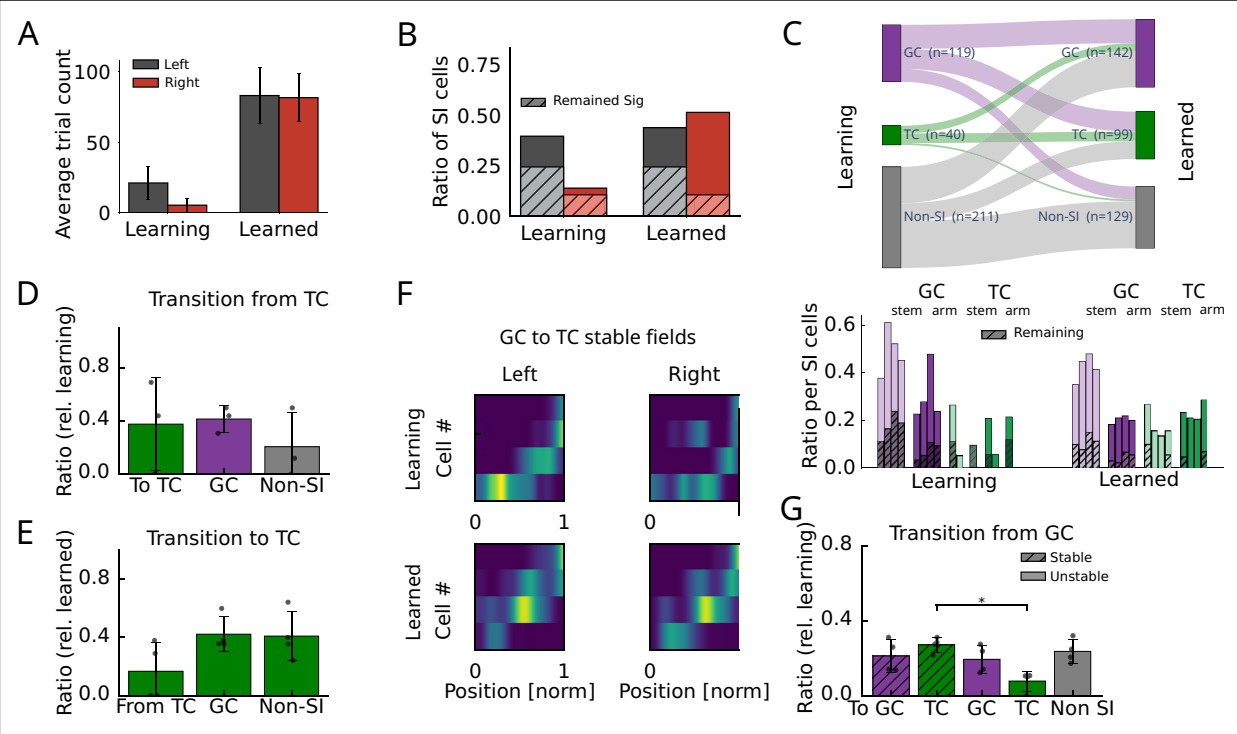

**Figure 2.** Development of task tuning during learning. (**A**) Trial counts (mean and standard deviation) over all animals in the learning and learned condition. Recordings in both conditions were limited to 15 minutes, explaining the lower number of trials in the learning group, where animals are still familiarized with the task structure. (**B**) Fraction of SI cell averaged across animals in the learning and learned condition. Hatched areas indicate proportions significant in both conditions (overlapping cells). (**C**) Top: transition between different cell types between learning and learned sessions. Bottom: amount of GCs and TCs normalized by the number of SI cells for all four animals (individual bars) during learning and learned condition. Hatched areas indicate proportions of cells that remain in the same category after learning. (**D**) Fractions of task cells (during learning) that transitioned to TC, GC, and non-SI cells after learning. (**E**) Fraction of task cells (for the learned condition) that arose from TC, GC, and non-SI cells during learning. (**F**) Example of GCs during learning (in one animal) that became TCs after learning. (**G**) Fractions of GCs (during learning) that transitioned to TC, GC, and non-SI cells with stable (hatched) and non-stable (plain) place field location. A Wilcoxon signed rank test showed significantly more (p=0.028, U=0, N=4 animals) stable TCs than non-stable TCs.

The online version of this article includes the following figure supplement(s) for figure 2:

**Figure supplement 1.** Task tuning per animal.

**Figure supplement 2.** Spatial selectivity during habituation is maintained in a subset of cells.

of the shuffle distribution ('Methods') were called generalized task phase selective cells (TCs), whereas SI cells for which the two maps were not significantly similar were called goal arm-selective cells (GCs). We found a ratio of about 48% of TCs among SI cells (*Figure 1F*) over all animals, considering only the 'learned' sessions in which the animals have reached a performance criterion described previously ('Methods'; *Muysers et al., 2024*).

In previous analyses, generalized task selectivity on the population level was shown to increase over learning *Muysers et al., 2025*; we therefore examined how goal arm- and task phase selectivity of SI cells develops over learning (*Figure 2*). We therefore subdivided sessions into early 'learning' sessions and late 'learned' sessions based on a behavioral performance criterion of 70% correct trials ('Methods'; *Figure 1—figure supplement 1A–C*). During learning sessions, animals had a strong bias towards left turns (*Figure 2A*, *Figure 2—figure supplement 1A*), which was also reflected in the over-proportionate number of SI cells with fields in leftward trials (*Figure 2B*, *Figure 2—figure supplement 1B and C*). After learning, both the behavioral and the neural biases vanished (*Figure 2A and B*). New SI cells were predominantly added for rightward trials (*Figure 2B*), whereas about a third of the SI cells for leftward trials did not remain SI in the learned condition (*Figure 2B*). We then distinguished learning-induced changes between goal arm and task phase encoding cells and found that in learning trials there was a strong bias towards goal arm encoding (*Figure 2C*), whereas in the learned

condition a substantial fraction of TCs (about 40%) was consistently found in all animals (*Figure 2C*). The increase in generalized task phase encoding on the level of SI cells matches the increase of task phase encoding previously observed on the population level in an extended population of animals (path-equivalent cells in *Muysers et al., 2025*). Despite the growth of the TC population, GCs remained to be the majority of cells, particularly those GCs with a field in the stem (*Figure 2C*). GCs on the stem have been previously referred to as splitter cells in the hippocampus (*Wood et al., 2000*) and suggested to encode episodic information. This is in line with our interpretation of GCs as goal encoding cells. Also, only about 40% of initial TCs remained TCs after learning (*Figure 2C and D*). The relative reduction in GCs, the increase in TCs, and the instability of a fraction of TCs over learning indicates that emerging TCs in the learned condition may have been generated from previous GCs. Indeed, ~40% of the TCs in the learned condition arose from GCs; however, also a similar fraction were not SIs during the learning condition (*Figure 2C and E*). For GCs that became TCs in the learned condition, the majority kept their place field (in an interval ±0.1 relative to the original field center; see example place cell population in *Figure 2F* and summary statistics in *Figure 2G*). Thus, task coding develops over learning by addition of existing SI cells with already formed place fields as well as the acquisition of TCs from previously non-spatially tuned cells. However, although most of the selectivity arises over learning, a small but significant fraction of cells show stable rate maps not only in comparison between learning and learned conditions (*Figure 2G*) but also taking into account the habituation phase, where the animals were exploring the behavioral arena without a task (*Figure 2—figure supplement 2*). A significant fraction of stable rate maps (habituation vs. learned) could particularly be observed in TCs but not in GCs (*Figure 2—figure supplement 2F*). This finding is consistent with the hypothesis of a limited preexisting scaffold.

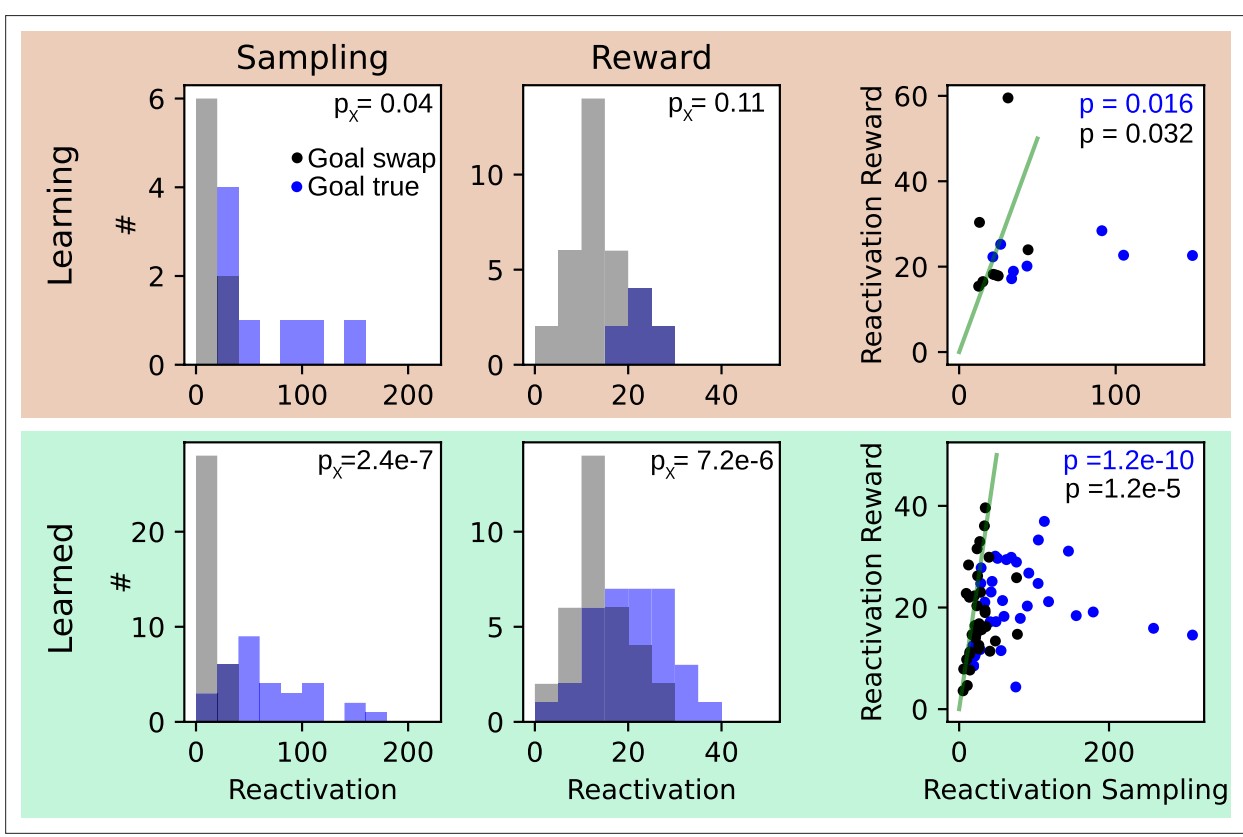

**Figure 3.** Reactivation of cofiring patterns is goal arm specific. Both, during the learning (top, orange) and the learned (bottom, green) condition, we computed the reactivation strength (*Peyrache et al., 2010*) of patterns observed during outward running for the sampling and the reward phase. For each session, reactivation was computed with respect to the true trial label (blue) and a swapped trial label (black). Left: for the true label reactivations were significantly ($p_X$) larger than for swapped labels, except for reward phase during learning. Right: reactivation during sampling was significantly ($p$) larger than during reward for true trial labels (identity is illustrated by green solid line). p-Values were derived from Wilcoxon signed rank tests. Statistics are given in the main text.

We next reasoned that the establishment of a task code on the single cell level during learning will likely be associated with changes in covariance structure on the network level. Since a subpopulation of TCs inherited their firing fields from previous place fields, we reasoned that these initial firing properties could reflect the activity of previously existing cell ensembles. In a first set of analyses, we were testing whether the rate covariance structure during the outward phase is 'reactivated' in the sampling and reward phases using the reactivation measure proposed in *Peyrache et al., 2009*; *Peyrache et al., 2010*; *Figure 3*. Both during the learning and learned phase reactivation during sampling was significantly larger than during the reward phase (Wilcoxon test, Learning: p=0.016, rank = 1, n=8 sessions, Learned: p=1.2e-10, rank = 0, n=34 sessions), arguing that already during sampling cell assemblies active during outward running start to co-fire. Moreover, comparing reactivation of left or rightward patterns during sampling and reward phases in correct trials between a true goal (patterns were taken from the trials with fitting goal arm) and a swap goal (patterns were taken from the trials with opposite goal arm) reveals significantly stronger reactivation in the sampling period for the 'true goal', both for learning and learned condition (Wilcoxon test; Learning: p=0.040, rank = 3, n=8 sessions; Learned: p=2.4e-7, rank = 30, n=34 sessions). These findings provide strong evidence that the cell assemblies during sampling are goal arm specific, that is, could underlie a sensory-driven recall initiated by odor sampling. For the reward phase, reactivation reached significant goal arm specific only in the learned condition (Wilcoxon test; Learning: p=0.11, rank = 6, n=8 sessions; Learned: p=7.2e-6, rank = 56, n=34 sessions). Reactivation was significantly larger during sampling vs. reward even in the swapped condition (Wilcoxon test, Learning: p=0.032, rank = 3, n=8 sessions, Learned: p=1.2e-5, rank = 42, n=34 sessions), and reactivation strengths were not significantly different between learning and learned condition (ranksum test; sampling: p=0.53, z=−0.64, n1=8, n2=34 sessions, reward: p=0.64, z=0.48, n1=8, n2=34 sessions; not shown in figure), suggesting that the co-firing structure reflects olfactory cue rather than task information (*Taxidis et al., 2020*; *Sun and Takehara-Nishiuchi, 2024*).

In addition to rate covariance (co-firing)-based assemblies, mnemonic processes have also been associated with temporal activity pattern, that is, activity sequences, particularly in the hippocampus.

We therefore next applied an unbiased sequence detection algorithm (*Chenani et al., 2019*) based on z-scored rank order correlations. In order to select only time points with many concurrently active cells, we set a relatively permissive threshold to the population calcium traces ('Methods') and extracted time intervals of 500 ms centered around the peaks of the population trace (*Figure 4A*). Burst rates were significantly larger during behavioral epochs (covering about 20–40% of the data thereby capturing a large amount of cofiring structure; *Figure 1—figure supplement 1D and E*) than during sleep and during periods of habituation to the arena (Task vs. sleep: p=3.1 × 10$^{-130}$, U=4,974,538, n=7390; task vs. arena: p=2.6 × 10$^{-5}$, U=1,546,011, n=6732; Mann–Whitney U rank tests; *Figure 4B*), already indicating limited opportunities for preplay (i.e., behaviorally correlated sequences occurring before the start of the outward phase). Moreover, burst rates during correct trials were significantly lower than during error trials (p=0.0014, U=3,461,962, n=6292; Mann–Whitney U rank test; *Figure 4B*). Ensembles were then identified according to the order of their center of mass calcium activity during population bursts (*Figure 4C*, *Figure 4—figure supplement 1A*). Hierarchical clustering of the z-scored correlations was done longitudinally over all behavioral sessions during the learning and learned conditions based on similarity scores computed for all pairs of bursts in these time frames ('Methods'). With this approach, we were able to identify repetitive activity sequences over the whole course of the experiment (see Rastermap plots; *Stringer et al., 2019 Figure 4D*, *Figure 4—figure supplement 2*). The method is particularly reliable in identifying sequences of little active cells, whereas it may also falsely detect some sequence clusters of highly active cells (active in more than 50% of bursts); see *Figure 4—figure supplement 1B*. Visual inspection of the Rastermap plots and correlating time points of sequence activations with behavioral trajectories exhibited large differences of sequence activation between learning and learned condition, both in the spatial patterns (example animal in *Figure 4D*) and the distribution of the sequences (*Figure 4D and E*). Rastermap plots (*Figure 4D*) also reveal little similarity of sequence expression between task and habituation or sleep condition. Sequence expression during the task was more variable between learning and learned condition than between correct and error trials (*Figure 4E and F*) as revealed from shuffled cluster identities (p-values were obtained using a permutation test with 200 shuffles: learning vs. learned, correct: p<0.005; failed: p<0.005; correct vs. failed, learning: p=0.8350; learned: p=0.415).

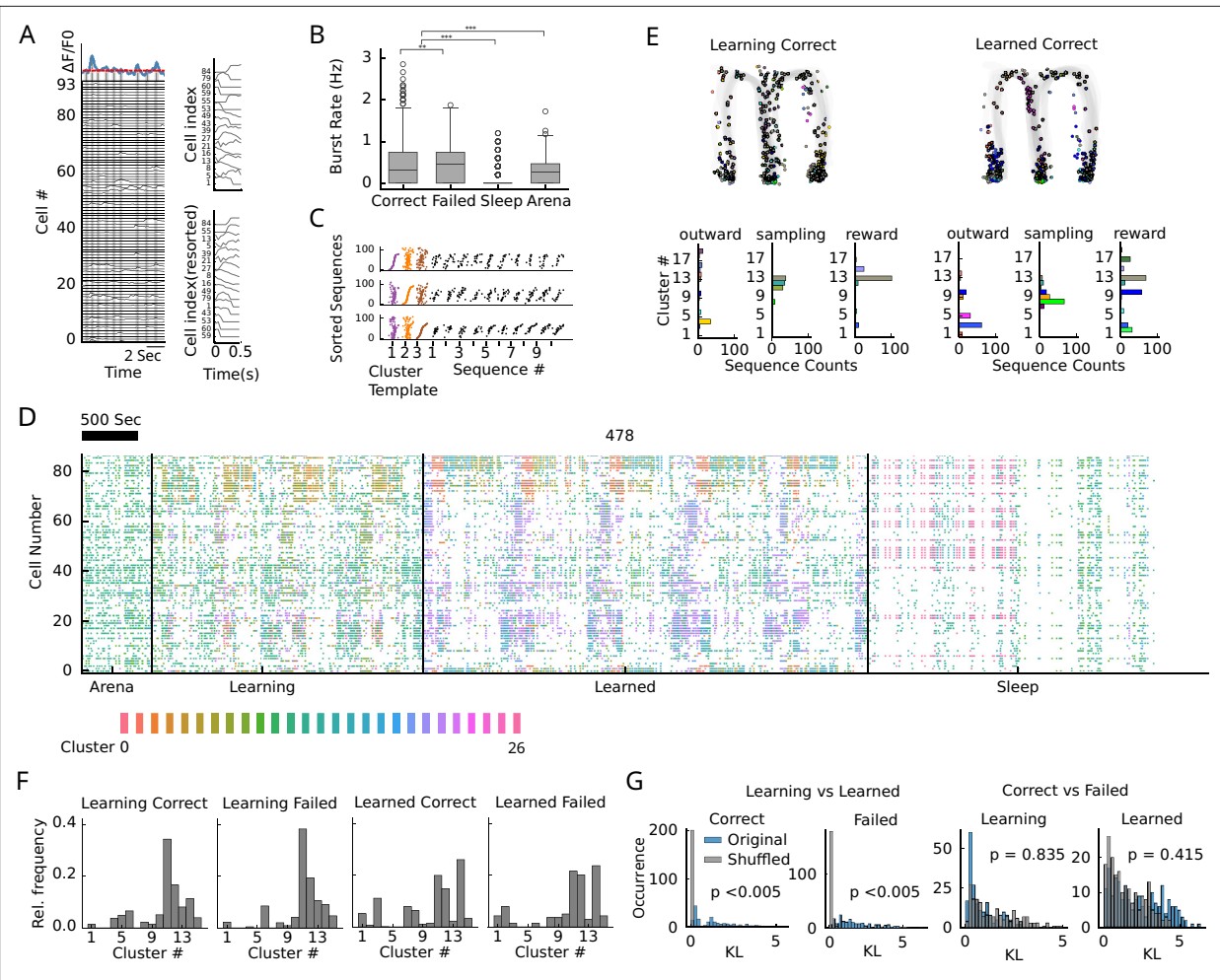

**Figure 4.** Recurring sequence motifs in one-photon recordings change with learning. (**A**) Example population recording of 93 cells. Blue trace indicates population (sum) activity. Significant peaks (above red line) of population activity are identified as bursts. Within 500 ms around burst peaks, cells are sorted according to the center of mass of the calcium trace. The index sorting is called a sequence. (**B**) Burst rates combined over all animals for different behavioral states. Correct and Failed correspond to Sampling, Outward, and Reward from *Figure 1A*, Sleep is derived from intermittent sleep epochs (see 'Methods'), Arena denotes the habituation period where the animal was left in the arena without task (see 'Methods'). Significance is derived from Mann–Whitney U rank tests (see main text). (**C**) Three example sequence cluster templates (colored) obtained via hierarchical clustering (see 'Methods') sorted according to first, second, and third cluster. Black sequences are representative cluster members of the cluster for which the cell index sorting was made in the panel. (**D**) Rastermap visualization (*Figure 4—figure supplement 2A*); and *Stringer et al., 2019* of sequences clusters (colors) for an example dataset from a single animal (478) for all sessions concatenated and sorted according to behavioral condition (Arena habituation, in task during Learning condition, in task during Learned condition, Sleep). Within each condition, activity is ordered according to sessions and within each session, intervals are ordered according to behavioral state (sampling, outward, reward, inward, right, left, respectively; see *Figure 4—figure supplement 2A* for illustration of ordering). Each point represents the time a cell had a calcium transient, with colors indicating the active sequence cluster. Sleep sessions were dispersed throughout but shown here as a concatenated block for visualization purposes. (**E**, top) Cluster identity (color) of a population burst as a function of position on the maze in the learning and learned condition. (**E**, bottom) Distributions of cluster identities for all bursts in the behavioral phases. (**F**) Relative frequencies of cluster identities of bursts in one example animal during learning and in the learned condition subdivided for correct and failed trials. (**G**) Kullback–Leibler (KL) divergence of cluster distributions (as in **F**) for all animals (blue) were obtained by subsampling the number of bursts in the learned conditions to match the number of bursts in the learning conditions (see 'Methods'). Animal-wise shuffles of learning and learned labels in the subsampled histograms are plotted in gray.

The online version of this article includes the following figure supplement(s) for figure 4:

**Figure supplement 1.** Validation of sequence clustering.

**Figure supplement 2.** Summary of raster maps.

Thus, our data suggest that task learning is associated with the establishment of a new robust population representation of current behavioral states.

In order to find functional correlates of the recurring sequence activity, we again computed SI values of sequence clusters and found a significant increase in SI clusters over learning (Wilcoxon signed rank test, p=0.02, W=1, n=8; *Figure 5A*, *Figure 5—figure supplement 1A and B*), similar to the increase in SI cells reported in *Figure 2B*. Separating goal arm-selective sequences (GS) from generalized task phase-selective sequences (TS) by applying the same approach as for GCs and TCs (see 'Methods'), we found an increase of both GSs and TSs over learning with a particular bias to goal specificity in the arm (*Figure 5B*), different from the stem bias of GCs reported in *Figure 2C*. The majority of sequence activity, however, was neither related to TSs nor to GSs (*Figure 5C*). The only exception was during the outward runs, where GSs and TSs together contributed to sequence activity by about as much as non-SI sequences. To understand the origin of goal and generalized task phase tuning of sequence clusters, we then examined the place fields of the contributing cells and found that particularly place fields in TCs had a stronger contribution to sequence activity in outward runs (*Figure 5D*). Moreover, to link place fields of SI cells to fields of SI sequence clusters, we computed distributions of distances between the peak of the cluster field and place fields of the contributing cells (pooled histogram shown in *Figure 5E*; histograms for individual animals in *Figure 5—figure supplement 1C*) and tested for significance, by comparing the distribution of peak distances between all cluster and cell fields. For all 7 detected TSs, the field centers of the TCs significantly determined task tuning of the sequence clusters, whereas for the 13 GSs, only less than half were determined by the place field centers of GCs (*Figure 5F*). Thus, there is a stronger link between TCs and TSs as compared to the link between GCs and GSs (p=0.0047; Fisher's exact test).

Taken together, our findings suggest that recurring sequences during outward runs are largely supported by TCs; however, the overall frequency of their occurrence, and potentially their functional relevance for task representation, is rather limited. We therefore were wondering whether sequence activity correlates with trajectory replay as demonstrated for the hippocampus (*Lee and Wilson, 2002*). By applying the Bayesian decoder generated for trajectory decoding from single cells (*Figure 1C*) trained on activity during population bursts (*Figure 5G*, *Figure 5—figure supplement 2* for examples), we found correlations of decoded position with time bin (within a 500 ms burst) strongly exceeding chance level only during outward and reward phase, for both GSs and TSs (*Figure 5H*), with only fractions of significant forward replays exceeding chance levels. We reckoned that strong correlations between decoded position and time during outward phases may simply reflect the behavioral sequence (running trajectory, as in the first example shown in *Figure 5G*). We therefore computed decoding errors during bursts in outward runs (*Figure 5I*) as root mean square error (RMSE) between true and decoded trajectories. The median RMSE of about 0.2 matches the RMSE from trajectory decoding in *Figure 1D*, supporting the interpretation that significant correlations between decoded position and time during SI sequences in outward runs in *Figure 5H* reflect the real-time task representation. Significant correlations of GSs and TSs in the reward phase, however, cannot relate to the current position of the animal and therefore have to be considered as replay. During the odor sampling phase, no such significant replay was observed for GSs and TSs, suggesting these bursts are not involved in odor sampling or even planning, but rather in the execution and evaluation of behavior. Non-SI sequence clusters showed small but significant bias to preplay in the sampling phase, indicative of Non-SI sequence clusters potentially containing a small contribution of spatial subsequences. However, in contrast to GSs and TSs that show significant proportions of forward replay (and not backward replay) in both the outward and reward phases, non-SI sequences during sampling did not reveal significant forward preplays exceeding chance expectations, despite the presence of recurring sequence motifs. Replay was not significantly observable during the reward phase in the learning condition (*Figure 5—figure supplement 1D*), which suggests that replay requires a stable task representation, or at least extensive exposure to the task. During the outward phase, however, correlations between decoded position and time were significant for all clusters (*Figure 5—figure supplement 1D*), even for the non-SI clusters, supporting the interpretation of a real-time behavioral representation even in the learning condition. Together with our finding of strong changes in sequence expression after learning (*Figure 4E*), these findings suggest that a representation of task develops during learning; however, it is not reflecting sequence structure during learning and habituation.

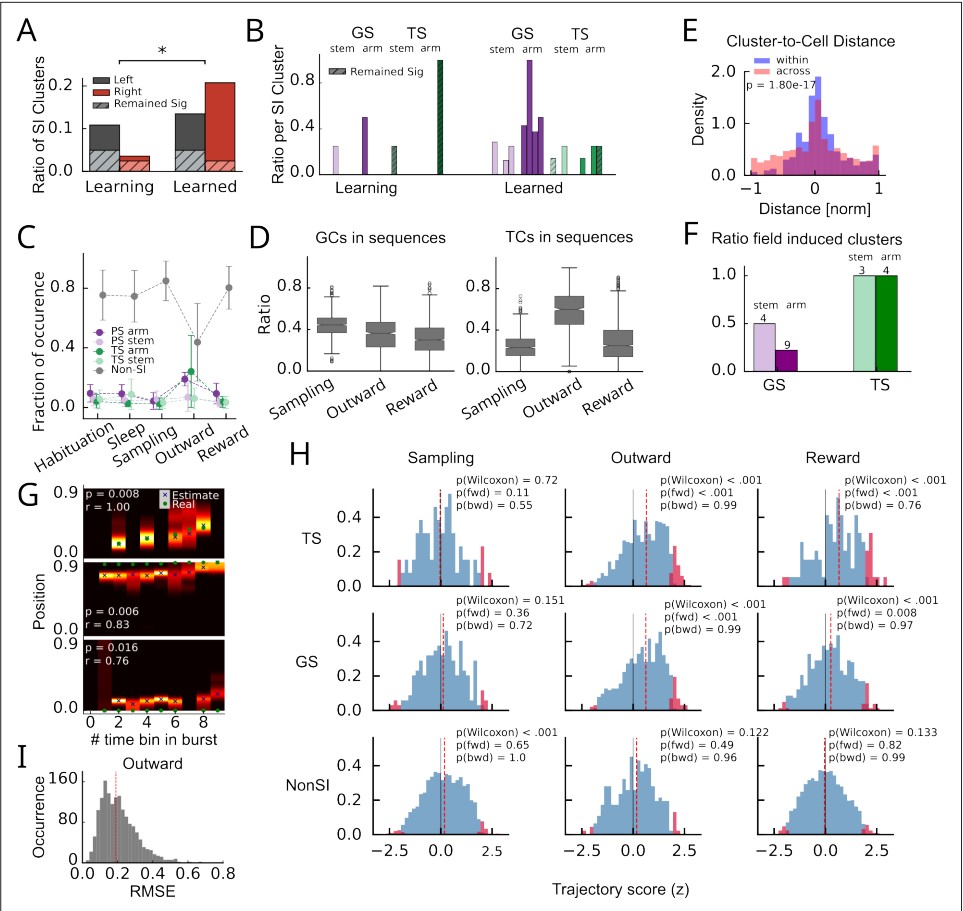

**Figure 5.** Generalized task phase- and goal arm-selective sequences are used for replay but not planning.
(**A**) Fraction of SI sequence clusters (relative to all clusters) combined across animals in the learning and learned condition. Hatched areas indicate proportions significant in both conditions. Significance is tested according to a Wilcoxon signed rank test (p=0.02). (**B**) Amount of GS and TS clusters normalized by the number of SI clusters for all four animals (individual bars) during learning and learned condition. Hatched areas indicate proportions of clusters remaining in the same category after learning. (**C**) Fraction of bursts attributed to GS, TS, and non-SI sequence clusters during different behavioral phases. The majority of bursts is not selective to behavioral parameters except in the outward phase, where about half of the bursts are associated with behavior. (**D**) Fractions of GCs and TCs contributing to bursts of the different cluster types and different behavioral phases. (**E**) Histogram of distances between field peaks of the cluster and the field peaks of the contributing cells for all SI clusters (blue). Distances between field peaks of all SI cells and all sequence clusters (across cluster) are shown in red. Clusters are called significantly cell induced if their absolute across-cluster distances are significantly larger than the within-cluster differences according to a one-sided Mann–Whitney U rank test. (**F**) Fraction of cell-induced clusters (see panel **E**) during the learned condition. Number on top of bars indicates total numbers of clusters of that type over all animals. (**G**) Examples of decoding during bursts. Posterior is color coded (max normalized). Crosses indicate maximum posterior, green dots the true position of the animal in the respective time bin (50 ms). Top: example during outward phase; middle: example during reward phase; bottom: example from odor sampling phase. (**H**) Distribution of z-scored rank order correlation coefficients between time bin and decoded position (trajectory scores) in each burst during sampling (left column), outward (middle column), and reward (right column) phase for generalized task phase-selective (top row), goal arm-selective (middle row), and non-SI (bottom row) clusters. Z-Scoring was performed using mean and standard deviation of a distribution obtained from randomly shuffling cell indices. Red bars indicate significant forward and backward replay for which trajectory score exceeded the upper 97.5 or the lower 2.5 percentile of the distribution generated by cell id randomization in the respective session. The fractions of forward and backward replay were then tested for significance above chance (0.025) according to a binomial test; p(fwd) and p(bwd). Red dashed lines indicate the medians, p(Wilcoxon) values are derived from a Wilcoxon signed rank test for zero median. (**I**) Distribution of RMSE from the outward phase over all animals (red line indicates median).

*Figure 5 continued on next page*

*Figure 5 continued*

The online version of this article includes the following figure supplement(s) for figure 5:

**Figure supplement 1.** Supplementary sequence analysis results.

**Figure supplement 2.** Burst decoding, additional examples.

To conclude, generalized task phase representation by TCs developed in parallel to a goal arm representation by GCs, where the latter is supported by an even larger number of SI neurons (predominantly in the stem of the M-maze, that is, splitter cells *Wood et al., 2000*) and an even larger amount of GSs (predominantly in the arm of the M-maze) than TSs. Thus, goal and task phase representation seem to coexist in rodent mPFC.

## Discussion

Longitudinal measurements of 1-p calcium activity in mice learning and performing an odor-guided navigation task allowed us to analyze the development of goal arm and generalized task phase selectivity on the single cell as well as on the population level. Recordings from animals that have not yet reached good task performance exhibited a preference for goal arm-selective neurons as compared to generalized task phase-selective neurons. However, also in task-proficient animals, with many generalizing TCs, a sizable amount of cells showed goal arm selectivity. GCs, however, largely remapped during learning, whereas TCs developed at least partly from initial SI cells with overlapping place fields, which indicates that those could be potential building blocks of task topology schemas. This hypothesis is further corroborated by a small but significant proportion of cells that exhibit a firing field at the same location in the arm even during the habituation phase, without performing a task potentially indicative of cross-condition generalization. Extending our analysis to the coordinated activity of neural sequences, we found that learning leads to the emergence of new structured activity patterns, indicative of refined network organization. A large number of recurring activity motifs were present, but TSs almost exclusively arose after learning. These changes were not significantly different between correct and failed trials and suggest a broad reconfiguration of sequence dynamics in the mPFC. These findings support the idea that task representations are generated de novo in the mPFC during learning.

Our analyses are based on a previously analyzed data set (*Muysers et al., 2024*; *Muysers et al., 2025*) and, using a complementary analysis approach, specifically confirm the main findings in *Muysers et al., 2025* regarding the higher stability and later development of TCs as compared to GCs. Our analyses extend the findings to habituation (no-task) periods, showing that a small fraction of TCs already has a firing field at the same position during habituation. Covariance and sequence-based analyses have not been performed in previous publications of this data set and show that the development of a task representation specifically aligns with the formation of fast sequences, as well as their replay during reward.

While memory-related fast (sub-second) sequence activity/temporally correlated firing has been previously reported in the PFC using electrophysiological methods (*Euston et al., 2007*; *Fujisawa et al., 2008*; *Peyrache et al., 2009*; *Kaefer et al., 2020*), the present study, to our knowledge, identified for the first time fast recurring neural sequence activity from 1p calcium data, based on correlation analysis, thereby allowing us to study the formation of fast sequences over a period of weeks. Using the center of mass approach for the calcium traces allows to interpolate between time samples and thereby limits the temporal resolution to the inverse of the sampling frequency rather than being constrained by the kinetics of the calcium indicator, at least for moderate sampling rates below about 50 Hertz.

So far, sequences on fast time scales have mostly been reported for the hippocampus during theta oscillations (*Foster and Wilson, 2007*) as well as during quiet wakefulness (*Diba and Buzsáki, 2007*; *Foster and Wilson, 2006*) and sleep *Lee and Wilson, 2002*; but see *Euston et al., 2007*; *Kaefer et al., 2020*. The functional role of hippocampal sequences is still debated and mostly assumed to relate to memory consolidation (*Girardeau et al., 2009*; *Fernández-Ruiz et al., 2019*) and planning (*Kay et al., 2020*; *Jadhav et al., 2012*). In contrast to hippocampal sequences, we find a very much reduced incidence rate outside the behavioral task, particularly during odor sampling. Odor sampling is a state of active sensing (*Martin et al., 2007*; *Kay, 2014*), and olfactory processing in PFC is not

only reflected by co-firing patterns (*Figure 3* and *Taxidis et al., 2020*; *Sun and Takehara-Nishiuchi, 2024*), but has also been reported to affect oscillatory activity (*Martin and Ravel, 2014*), particularly phase amplitude coupling of beta and gamma oscillations to the theta rhythm (*Ramirez-Gordillo et al., 2022*). The absence of preplay during the sampling phase is, thus, likely reflecting that active sensing and planning are distinct behavioral states. The absence of replay, and the generally low levels of sequence activity, during sleep/rest states suggests that they do not directly echo enhanced hippocampal sequence activity during sleep, questioning their role in consolidation. During outward runs, sequences seem to reflect the current behavioral state of the animal in the task and in space, corroborating that mPFC encodes task phase not only on the single cell level but also on the population level (*Muysers et al., 2025*). These outward sequences could, thus, simply arise from trials with high firing rates. More interestingly, though, is our finding on replay of GSs and TSs in the reward area. In contrast to the hippocampus where sequences of place cell activity are expressed in reversed order at the end of the track but in forward order at the beginning of the track (*Diba and Buzsáki, 2007*; *Foster and Wilson, 2006*), we found exclusively forward replay (*Figure 5H*). Together with our finding of increased burst activity during error trials, this suggests that sub-second mPFC sequences may be associated with the evaluation of behavior. In addition to GSs and TSs, we found that most of the recurring sequences are not related to behavior (not SI), even during outward runs. Ongoing mPFC activity, hence, may reflect latent variables beyond experimental control.

The distinctiveness of hippocampal and PFC sequences has been emphasized previously (*Kaefer et al., 2020*; *Nardin et al., 2023*). It is broadly assumed that mPFC sequences positively correlate with behavioral performance (*Kaefer et al., 2020*; *Tang et al., 2021*) – consistent with our observed establishment of stable task sequences after learning – and that they are associated with trajectory reactivations (*Tang et al., 2021*; *Nardin et al., 2023*), which is consistent with our finding of enhanced trajectory indices in the reward area (*Figure 5H*). However, the suggestion that mPFC sequences may also support planning (*Tang et al., 2021*) could not be confirmed by our work as sequences in the odor sampling phase were absent (*Figure 5H*). The difficulty of interpreting planning-associated activity may also partly result from the many activity bursts that are not task-related (*Figure 5C*) and the relatively smaller amount of GCs and TCs being active in the odor sampling phase during population bursts (*Figure 5D*). Moreover, planning-related sequences may be masked by activity related to active sensing. Additionally, the 1p calcium signal may simply not be able to detect planning-associated activity in case it relies only on weak activity modulations.

In cognitive neurosciences, the prefrontal cortex is often assumed to support flexible behavior by schemas, that is, prototypic behavioral patterns that can be applied under variable conditions that share the same important contextual aspect. PFC neural activity in rodents has also been linked to schemas when task structure remained stable across spatial locations (*El-Gaby et al., 2024*). The use of schema-like activity during learning of a novel task, however, is only little investigated. A small sample of cells has previously been identified to maintain task structure during rule switching (*Reinert et al., 2021*). While the limited subset of GCs we found to translate their place fields into a generalized task field over learning could be interpreted as potential building blocks of schema of the spatial environment, task-related activity mostly developed independently from previous spatial representations. We thus hypothesize that in order to transfer a larger degree of schema-related representations to a new task, high behavioral similarity between the tasks is required.

## Methods

Experimental data was mostly published previously in *Muysers et al., 2024* and *Muysers et al., 2025*. Data derived from periods of sleep and habituation to the arena are previously unpublished. All experiments were performed in agreement with national legislation (licenses G18-145 and G19-145 approved by the Regierungspräsidium Freiburg). A detailed description of experimental methods can be found in the original publication. The next paragraph only provides a brief summary.

### Experimental methods and data preprocessing

Data was derived from four Thy1-GcaMP6f mice (Jackson Labs #025393; two females and two males). Animals were 11 weeks (75–78 days) old upon lens implantation and 16 weeks upon the beginning of the handling. The analyzed animals represented the subset of the previously published animals

(*Muysers et al., 2024*; *Muysers et al., 2025*) for which sufficiently many cells could be tracked throughout the experiment (including sleep and arena) to perform longitudinal sequence clustering.

Animals were first habituated to the experimenter and the experimental room for at least 3 days. The animals were then exposed to the behavioral arena (M-shaped maze) for 3 days (habituation to arena). The two different odors (vanilla; coconut) and the reward were introduced subsequently, and the animals were exposed for a week to 'forced' trials: the odor was presented at the center arm of the arena and only the correct door to the respective side arm was opened, a reward was consequently given at the end. Thereafter, the 'free choice' trials started, where the odor was presented at the center and both side-arms were accessible for free choice. Animals were trained for 2 months. In this time, all four mice learned the task (≥70% correct across three consecutive days).

Behavioral epochs (sampling, center, side, reward, all for left and right respectively) of the task were extracted, based on xy-thresholds of arena locations obtained from camera tracking.

Integrated GRIN lenses of ø 1 mm (Inscopix #1050-004637) were used to gain optical access to the prefrontal cortex via a ø~1 mm craniotomy above the mPFC (centering on 1.7 mm anterior and 0.6 mm lateral of bregma).

Neuronal activity was recorded when the animals were first exposed to the behavioral arena and throughout all training sessions. For analysis, we focused on the recordings of 3 days freely moving in the arena, 4 days of learning (~50% correct trials) and 4 days after reaching the criterion ('earned'). Additionally, animals were recorded during sleep in their homecage before and after learning of the task.

The open-source Python toolbox Caiman was used to identify neurons in the recorded one-photon videos. Extracted components were manually checked (https://github.com/chenhungling/CaimanGUI; *Chen, 2023*) before further analysis.

Calcium traces, as extracted with Caiman, were first corrected for slow drifts using a running percentile filter (10th percentile, 30 s window). The mean (baseline) and standard deviation (σ) were then calculated iteratively. In each iteration, signal above 3σ was excluded and baseline and σ were calculated again until the relative change in σ was smaller than 0.1%. The traces were baseline-subtracted and normalized by the respective σ.

Significant transients were detected by keeping the signal >3σ and a minimum duration of 0.2 s. The rest of the signal was set to 0. CellReg was used to track cells across sessions (*Sheintuch et al., 2017*).

## Reactivation analysis

We followed *Peyrache et al., 2010* and computed reactivation strength $R_l$ of a population pattern $b_l$ evoked by z-scored activity population vectors $z$ (during sampling or reward) according to $R_l = (b_l^T z)^2$, with $b_l$ denoting the $l$th significant eigenvector of the $n \times n$ covariance matrix of the z-scored activity vectors during the outward (task) phase (with $n$ denoting the number of simultaneously recorded neurons). Time binning was extended to 250 ms and only those sessions were used in which the total duration of outward phase for each trial type (left/right) was exceeding 12.5 s. Significance of eigenvectors was assessed via the Marchenko–Pastur bound $(1+(n/T)^{1/2})^2$, with $T$ denoting the number time samples. Total reactivation $R$ was then calculated as the weighted sum of all time-averaged $R_l$, weighted with the corresponding eigenvalue of pattern $b_l$.

## Burst detection

The population activity for each recording session was calculated by summing the neuronal activity across neurons at each time point. Subsequently, for each animal, the population activity from all sessions was concatenated to estimate the overall activity.

Burst events were identified when population activity exceeded a threshold of 0.5 standard deviations (σ). Neural activity from active cells within a 500 ms window, centered on each burst, was extracted, and cells were sorted based on the center of mass time point of the calcium trace. For each population burst, the sorted indices of these cells were compiled into individual sequence vectors. Only sequences comprising at least five cells were included in subsequent analyses. Using the center of mass inherently normalizes for heterogeneous firing rates among neurons, a factor that can lead to spurious sequence order in electrophysiological recording, because high firing neurons would always fire earlier.

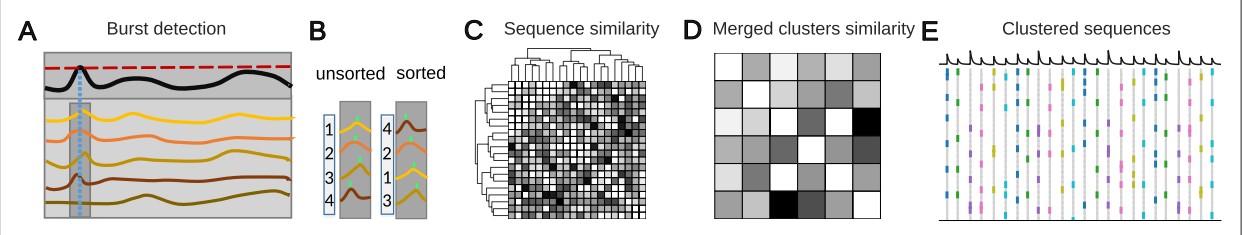

**Figure 6.** Schematized workflow of sequence detection. (**A**) For five example cells (calcium traces in colored lines) are summed to yield the population activity (black line). Whenever the population trace exceeds a threshold (red dashed line), a 500 ms window around the peak is considered a population burst (dark gray rectangle). (**B**) Single-cell activity within the 500 ms window is used to calculate center of mass for every cell. For every burst, cells are reordered according to their center of mass. The resulting ordered indices are referred to as 'sequence'. (**C**) A sequence similarity matrix is computed from the z-scored Spearman correlation between two sequences according to *Chenani et al., 2019*; The binarized matrix (1: significant z-value; 0: non-significant z-value) is subject to hierarchical clustering. (**D**) Similarity z-scores are again computed for all pairs of cluster templates (mean sequences in a cluster) and the clustering is iteratively repeated until merging criteria are no longer fulfilled. (**E**) Sorted sequences, with colors indicating cluster membership.

## Clustering sequences

To quantify the similarity between sequence pairs, Spearman's rank correlation coefficients were calculated based on the ordinal ranks of neuron activations in each pair of bursts. Only the L neurons active in both sequences of a pair were taken into account. By construction, Spearman's correlations are symmetric. In order to check for significance of sequence similarity, we followed the approach detailed in *Chenani et al., 2019*: correlation values were z-scored according to the distribution of correlation coefficients derived from random sequences of same length L (i.e., all permutations of the sequence [0,1,2,…,L]). Only pairs with significant z-scores were considered further. The resulting binary matrix (1: significant; 0: nonsignificant) was then used for agglomerative hierarchical clustering with Ward's linkage method, producing discrete sequence clusters.

## Iterative merging of cluster templates

After initial clustering of sequences and construction of a template for each cluster by averaging its member bursts, we quantified cluster similarity using the procedure described above for individual sequences. At each iteration, we identified the cluster pairs with significant similarity and merged their underlying members if they were too similar, that is, the z-score of a template pair exceeded a value of 2.5. Following each merge, cluster templates and similarity matrix were recomputed, and the procedure was repeated until no pair of clusters satisfied the merging criterion. The workflow for cluster identification is summarized in *Figure 6*.

## Occupancy

Occupancy was calculated by first generating a histogram of the animal's position. The counts in each bin of the histogram were multiplied by the time spent in the corresponding bin, converting position counts into occupancy values per bin. To smooth the occupancy data, a Gaussian filter with sigma = 5 cm was applied.

## Rate map

The smoothed transient histogram was divided by the occupancy to obtain the firing rates for each position bin.

## Spatial information

Spatial information was calculated using the rate maps of neurons to identify place cells:

$$I = \sum_{x} P(x) \cdot r(x) \cdot \log_2\big((r(x) + \varepsilon)/(r_0 + \varepsilon)\big)$$

where *r(x)* is the firing rate in a spatial bin *x*, *P(x)* is the probability of the animal being in bin *x*, $r_0$ is the overall mean firing rate, and $\epsilon$ is a small constant (1e-10) to prevent division by zero.

To identify place cells with significant spatial information, we performed circular shuffling of transient times within each trial period and calculated the spatial information for the shuffled rate maps. This process was repeated 500 times, generating 500 shuffled datasets. The spatial information of the original data was then compared to the shuffled distributions. Neurons with spatial information significantly greater than the shuffled data were classified as spatially informative (SI) cells.

### Bayesian decoder

A Bayesian decoder was employed to calculate the posterior probabilities of all position bins. This method assumes that transient counts follow a Poisson distribution and combines transient counts and rates to compute the likelihood of the animal being at each position bin, given by

$$\log L(y) = \sum_{i=1}^{N} [k_i \log(r_i(y) \cdot \Delta t) - r_i(y) \cdot \Delta t - \log(k_i!)]$$

where $L(y)$ is the likelihood of the animal being at position $y$, $N$ is the total number of neurons, $k_i$ is the count of transients of neuron $i$, $r_i(y)$ is the mean transient rate of neuron $i$ at position $y$, and $\Delta t$ is the decoding time bin.

The posterior probability $P(y)$ is obtained by normalizing the likelihoods across all position bins:

$$P(y) = L(y) / \sum_{y'} L(y')$$

The estimated position of the animal is determined as the position $\hat{y}$ that maximizes the posterior probability $\hat{y} = \text{argmax}_y P(y)$.

### Position estimation

To estimate the position of the animal, peaks of transient calcium traces were detected for each neuron. Peak detection was performed using the find_peaks function from Python's SciPy library, with the parameters set to height = 0, width = 50 ms, and distance = 100ms. The height = 0 parameter ensured that all peaks above zero were considered, width specified the minimum width of peaks to be detected, and distance set the minimum required distance between consecutive peaks to avoid detecting closely spaced events as separate peaks.

A Bayesian decoder was trained using the detected peaks that were not part of bursts or their neighboring points. The decoder was then tested on the time points where bursts occurred to estimate animal position.

### Estimate position using burst data

To estimate the position of the animal using burst signals of sequences, a Bayesian decoder was trained using rate maps from outward runs as the training data. Time points when sequences were detected were excluded from the training data to ensure that the decoder's training was not influenced by these events. The decoder was then tested exclusively on the 500 ms burst signals centered around the detected sequences. Event times of bursts during the sampling, outward, and reward periods were used as test data. Animal position during these times was estimated by taking the maximum likelihood of the posterior probability distribution provided by the Bayesian decoder.

### Subsampling

To account for the unequal number of trials between the learning and learned conditions, the number of sequences in the learned condition was equalized by subsampling. The learned data were subsampled to match the number of sequences in the learning condition. This process was repeated 50 times to ensure robustness. Clustering was then performed independently on each of the 50 subsampled datasets.

### Kullback–Leibler (KL) divergence

To quantify the differences in cluster distributions between the learning and learned conditions, the KL divergence was applied. The KL divergence is defined as $DKL(P \parallel Q) = \sum_i P(i) \log(P(i)/Q(i))$, where $P(i)$

and $Q(i)$ represent the probabilities of sequence cluster $i$ during the learning and learned conditions, respectively.

The KL divergence was calculated for each of the 50 subsampled datasets, resulting in a distribution of KL values derived from the original data. To assess significance, a null distribution was generated by randomly assigning sequences to the learning and learned conditions and computing the KL divergence for 50 such shuffled datasets. The KL divergence from the original data was considered significant if it exceeded the 95th percentile of the null distribution.

### Assigning sleep and arena sequences to clusters

Assignment of sleep, arena, and task sequences to clusters was performed by calculating the rank order correlation between each sequence and all cluster templates. Each sequence was then assigned to the cluster with the highest similarity score based on the computed correlations.

### Cluster-to-cell distance

The distances between cells and clusters were calculated by measuring the differences between the peak positions of the cells' rate maps and the sequence cluster templates. Distances were histogrammed for two conditions: either cell and sequence cluster are of same type (task arm, task stem, place arm, place stem) or including all cells. The two resulting distributions of distances were compared using a two-sided Mann–Whitney U test.

### Task cells and goal arm cells

To classify SI cells as goal arm-selective cells or generalized task phase-selective cells based on their firing patterns, rate maps during both left and right runs were analyzed. Cells were first sorted according to their rate maps from left runs. The analysis was then repeated, sorting cells according to their rate maps from the right run. For both sorting conditions, correlation coefficients were computed between the sorted rate maps of left and right runs for each region. To assess significance, a null distribution was generated by computing correlations between the rate maps of one condition and cell-index shuffled rate maps of the other condition. Cells with correlation coefficients exceeding the 95th percentile of this null distribution were classified as significant TCs, indicating similar activity during left and right runs. Conversely, cells with lower correlations were classified as GCs.

The rate maps were divided into two regions: the stem (linearized position 0–0.5) and the arms (0.5–1). Hence, the analysis identified four cell types: GCs in the arms (GC-arm), GCs in the stem (GC-stem), task cells in the arms (TC-arm), and task cells in the stem (TC-stem).

### Task sequences and goal arm sequences

Criteria for defining GCs and TCs were also applied to categorize clusters based on their firing patterns and spatial activity into goal arm-selective sequence clusters (GS) and generalized task phase sequence clusters (TS): firing rate maps were computed for each cluster. Only clusters were further considered for which at least one of the firing rate maps (right/left) had a significant SI. Rate maps were then correlated between left and right trials. If the correlation coefficient exceeded the 95th percentile of a cell index, shuffling the clusters was considered a TS, otherwise a GS. The analysis was restricted to clusters containing at least five sequences.

### Stability of place fields

To assess the stability of firing fields after learning, we compared the rate maps of SI cells between learning and learned conditions. For each cell, the rate map of one condition was circularly shifted, and the correlation between the shifted and original maps was computed. If the maximum correlation occurred within a shift range of ± 0.1, the cell was considered stable.

## Acknowledgements

This work was funded by the German Research Association (DFG) under grant numbers LE2250/20-1 (C.L., FOR5159), BA 1582/16-1 (M.B. FOR5159), BA 1582/24-1 (M.B.), and SA 3609/2-1 (J-F.S.FOR 5159).

## Additional information

### Funding

| Funder | Grant reference number | Author |
|---|---|---|
| Deutsche Forschungsgemeinschaft | LE2250/20-1 | Christian Leibold |
| Deutsche Forschungsgemeinschaft | BA 1582/16-1 | Marlene Bartos |
| Deutsche Forschungsgemeinschaft | BA 1582/24-1 | Marlene Bartos |
| Deutsche Forschungsgemeinschaft | SA 3609/2-1 | Jonas-Frederic Sauer |

The funders had no role in study design, data collection and interpretation, or the decision to submit the work for publication.

### Author contributions

Hamed Shabani, Conceptualization, Data curation, Software, Formal analysis, Investigation, Visualization, Methodology, Writing – original draft, Writing – review and editing; Hannah Muysers, Data curation, Methodology, Writing – review and editing; Yuk-Hoi Yiu, Resources, Software, Methodology; Jonas-Frederic Sauer, Marlene Bartos, Resources, Supervision, Funding acquisition, Writing – review and editing; Christian Leibold, Conceptualization, Formal analysis, Supervision, Funding acquisition, Investigation, Methodology, Writing – original draft, Project administration, Writing – review and editing

### Author ORCIDs

Hamed Shabani https://orcid.org/0000-0003-2057-1364
Hannah Muysers https://orcid.org/0000-0002-6883-2706
Yuk-Hoi Yiu https://orcid.org/0000-0002-1997-9277
Jonas-Frederic Sauer https://orcid.org/0000-0002-6854-7294
Marlene Bartos https://orcid.org/0000-0001-9741-1946
Christian Leibold https://orcid.org/0000-0002-4859-8000

### Ethics

Data was obtained from animal experiments conducted in agreement with national legislation (licenses G18-145 and G19-145 approved by the Regierungspräsidium Freiburg).

Reviewer #1 (Public review): https://doi.org/10.7554/eLife.106981.4.sa1
Author response https://doi.org/10.7554/eLife.106981.4.sa2

## Additional files

### Supplementary files

MDAR checklist

### Data availability

Analysis code is available from the GitHub-Repository https://github.com/HamedShabani/PFC_sequence_analysis, (copy archived at *Shabani, 2025*). The datasets that went into the analyses are available under https://doi.org/10.12751/g-node.b8k1yn. Part of the data is overlapping with the repository of a previous publication (https://doi.org/10.5281/zenodo.10528243).

The following datasets were generated:

| Author(s) | Year | Dataset title | Dataset URL | Database and Identifier |
|---|---|---|---|---|
| Shabani H, Muysers H, Yiu YH, Sauer JF, Bartos M, Leibold C | 2025 | Formation of task representations and replay in mouse medial prefrontal cortex dataset | https://doi.gin.g-node.org/10.12751/g-node.b8k1yn/ | G-Node GIN, 10.12751/g-node.b8k1yn |
| Muysers H, Chen HL, Hahn J, Folschweiller S, Sigurdsson T, Sauer JF, Bartos M | 2024 | Code & Data for 'A persistent prefrontal reference frame across time and task rules' | https://doi.org/10.5281/zenodo.10528244 | Zenodo, 10.5281/zenodo.10528244 |

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
