## [Editor Report · eLife Assessment]

This **useful** study characterizes the evolution of medial prefrontal cortex activity during the learning of an odor-based choice task. The evidence provided is **solid**, providing quantification of functional classes of cells over the course of learning using the longitudinal calcium recordings in prefrontal cortex, and quantification of prefrontal sequences. However, the experimental design appears to provide limited evidence to support strong conclusions regarding the functional relevance of neural sequences. The study will be of interest to neuroscientists investigating learning and decision-making processes.

---

## [Referee Report · Reviewer #1 (Public review)]

This study presents a useful finding about development of task representations in mouse medial prefrontal cortex using 1-photon calcium recordings in an olfactory-guided spatial memory task. A key strength of the study is the use of longitudinal recordings allowing identification of task-related activity that emerges after learning. The study also reports existence of neuronal sequences during learning and their replay at reward locations. The evidence provided is solid, providing quantification of functional classes of cells over the course of learning using the longitudinal calcium recordings in prefrontal cortex, and quantification of prefrontal sequences.

(1) The authors continue to state that task phase selective cells (non-splitter) cells can be considered as "cross-condition generalization" and interpret them as "potential building blocks of schemas". However, cross-condition generalization requires demonstration of cross-condition generalization performance (CCGP) of neural decoders across task conditions, which is not shown here.

(2) The authors note that correlations on short time scales are not similar between sampling and reward phase, acknowledging that these two represent different behavioral states in a cued-memory task, and that the manuscript should more clearly distinguish replay with "pure sequences". However, while the last line in the abstract states that "sub-second neural sequences in the mPFC are more likely involved in behavioral outcomes rather than planning future actions", references are made throughout the manuscript to preplay/replay sequences, including results primarily for non-cued spatial memory tasks, in which there is no cued sampling phase. For example, lines 259-263 state "During odor sampling phase, no such significant replay was observed..." and "... sequence clusters showed small but significant bias to preplay in the sampling phase". If the authors want to distinguish between replay and "pure" sequences, then the terminology "replay" and "preplay" should not be used here.

Further, large parts of the Discussion are devoted to comparison to hippocampal ripple-associated replay. Lines 355-356 in Discussion state that "the suggestion that mPFC sequences may also support planning [Tang et al., 2021] could not be confirmed by our work as sequences in the odor sampling phase were absent". It should be clarified that this is a comparison between what the authors term "pure sequences" in the sampling phase of an odor-cued task, and internally generated sequences during hippocampal ripples in a non-cued spatial memory task, so this is not a like-for-like comparison.

---

## [Author Response]

The following is the authors’ response to the previous reviews

**Public Reviews:**

**Reviewer #1 (Public review):**
There are a few remaining issues:(1) The manuscript quantifies changes over learning in prefrontal goal-selective cells (equated to "splitter" place cells in hippocampus) and task-phase selective cells (similar to non-splitter place cells that are not goal modulated). A subset of these task cells remain stable throughout learning, and are equated to schema representations in the study. In the memory literature, schemas are generally described as relational networks of abstract and generalized information, that enable adapting to novel context and inference by enabling retrieval of related information from previous contexts. The task-phase selective cells that stay stable throughout learning clearly will have a role in organizing task representations, but to this reviewer, denoting them as forming a schema is an unwarranted interpretation. By this definition, hippocampal non-splitter place cells that emerge early in learning and are stable over days would also form a schema. Therefore, schema notation cannot just be based on stability, it requires further evidence of abstraction such as cross-condition generalization.

We agree with the reviewer that task phase selective cells (“non-splitter cells”) alone do not fulfill the “relationality” criterion of schemas. We found only few of them, and so we cannot really say something about how they covary. We, however, would like to stress that our finding that task phase selective cells have stable firing field comparing learned (task) and habituation (no-task) conditions can be considered as “cross-condition generalization.” We have further specified our discussion of schemas with a particular emphasis on a potential interpretation of the generalizing task phase cells as “potential building blocks of schemas.”

(2) The quantification of prefrontal replay sequences during reward is useful, but it is still unconvincing that the distinction between existence of sequences in the odor sampling phase and reward phase is not trivially expected based on prior literature. This is odor guided task, not a spatial exploration task with no cues, and it is very well-established (as noted in citations in the previous review) that during odor sampling, animals' will sniff in an exploratory stage, resulting in strong beta and respiratory rhythms in prefrontal cortex. Not having LFP recordings in this task does not preclude considering prior literature that clearly shows that odor sampling results in a unique internal state network state, when animals are retrieving the odor-associated goal, vastly different from a reward sampling phase. The authors argue that this is not trivial since they see some sequences during sampling, although they also argue the opposite in response to a question from Reviewer 2 about shuffling controls for sequences, that 'not' seeing these sequences in the sampling phase is an internal control. The bigger issue here is equating these sequences during sampling to replay/ preplay or reactivation sequences similar to the reward phase, since the prefrontal network dynamics are engaged in odor-driven retrieval of associated goals during sampling, as has been shown in previous studies.

We agree with the reviewer that sampling and reward phase represent two very different behavioral states. Nevertheless, correlations on short time scales could be similar, which we show is not the case and therefore we do not consider this result trivial. Regarding the interpretation of sequences, we apologize that we have not been sufficiently clear on distinguishing replay with pure sequences. While we find such sequences in the sampling phase (indicative of fast temporal correlation structure beyond cofiring quantified in Figure 3) they are NOT pre/replaying any task related information. Otherwise, our results are fully in line with previous literature on oscillations that we have included in the previous round of revisions. We added a similar explanation at multiple instances in the Results and Discussion section.

**Reviewer #2 (Public review):**
Comments on revisions:Further changes are needed to improve the description of the methods and the discussion needs to be extended to contrast the results with previously published results of the group. Some control figures would also be needed to quantitatively demonstrate, across the entire dataset, that sequence detection did not identify random events as sequences, even if the detection method was designed to exclude such sequences. For example, showing that sequences are not detected in randomised data with the current method would better convince readers of the method's validity.

We have added control quantifications from time randomized sequences which produce a much lower amount of detected sequences. See response below.

Although differences in the classification scheme relative to the Muysers et al. (2025) paper have been explained, the similarity (perhaps equivalence of results) is not sufficiently acknowledged - e.g., at the beginning of the discussion.

We have added a paragraph at the beginning of the Discussion on how our results align with the Muysers et al. 2025 paper.

Although the control of spurious sequences may have been built into the method, this is not sufficiently explained in the method. It is also not clear what kind of randomization was performed. Importantly, I do not see a quantification that shows that the detected sequences are significantly better than the sequence quality measure on randomized events. Or that randomized data do not lead to sequence clusters.

In response to this question, we have added the requested shuffling control (Supplement 1B to Figure 4). In the shuffled data the amount of detected recurring sequence clusters is only about half of those in the original data. The amount of bursts assigned to clusters in the shuffled data only remains 46% of the originally assigned bursts on average, clearly indicating that the detected sequences in the non-randomized data cannot be explained without assuming stable temporal order.

Some clusters, however, are still detected in randomized data, which, however, is expected if participation of cells is heterogeneous with some highly active cells occurring in more than half of the bursts. Then random sequences spuriously occur above chance level representing the clusters of random order of few highly active cells. In line with this interpretation, we see that

(1) Bursts that were removed after shuffling have exactly 0 high-firing cells

(2) Clusters derived from shuffled sequence have a less sparse contribution of high firing cells, i.e., high firing cells contribute to significantly more clusters in randomized data than in nonrandomized data.

The difference in the distribution of high firing cells further indicates that sequences obtained with and without randomization are of different quality.

The spurious (false positive) clusters detected after randomization nevertheless may have a physiological meaning as they identify rate coactivation patterns that were also picked up by analysis in Figure 3.

Also, it is still not clear how the number of clusters was established. I understand that the previously published paper may have covered these questions; these should be explained here as well.

The Methods sections states “The [cluster merging] procedure was repeated until no pair [of clusters] satisfied the merging criterion.”

Also, the sequence similarity description is still confusing in the method; please correct this sentence "Only the l neurons active in both sequences of a pair were taken into account."

We do not see what is wrong with this sentence. To avoid confusion.” we have replaced lower case l with upper case L as sequence length.

**Reviewer #3 (Public review):**
One comment is that the threshold for extracting burst events (0.5 standard deviations, presumably above the mean) seems lower than what one usually sees as a threshold for population burst detection, and the authors show (in Supplementary Fig 1) that this means bursts cover ~20-40% of the data. However, it is potentially a strength of this work that their results are found by using this more permissive threshold.

We have added further specifications following the Reviewer’s suggestion and now mention that the threshold is permissive and “capturing large amount cofiring structure.”